# Interference Signal Suppression Algorithm Based on CNN-LSTM Model

**DOI:** 10.3390/s25165048

**Published:** 2025-08-14

**Authors:** Ningbo Xiao, Zuxun Song

**Affiliations:** School of Electronics and Information, Northwestern Polytechnical University, Xi’an 710072, China; zuxun_10086@163.com

**Keywords:** interference signal, CNN-LSTM, suppression algorithm

## Abstract

Sensors and anti-interference technology have a complementary relationship. The anti-interference capability directly affects the measurement accuracy, reliability, and stability of sensors. In complex electromagnetic or natural environments, sensors are inevitably influenced by various interference sources. Effective anti-interference technology is the key to ensuring the normal operation of sensors, and suppressing interference signals is one of the key links to improving communication quality. This paper proposes a CNN-LSTM-based interference signal suppression algorithm, aiming to enhance the anti-interference capability of wireless communication systems through deep learning technology. The algorithm utilizes CNN to extract the spatial features of the signal and LSTM to capture the temporal dynamic characteristics of the signal, outputting a predicted signal to effectively suppress interference signals. The performance of the experimental simulation algorithm under different interference scenarios was evaluated and compared with three models: LSTM, BO-LSTM, and CNN-GRU. The results demonstrated that this algorithm had a small error and a high degree of regression fitting. Finally, the effectiveness of the algorithm was verified by using the signal propagation model based on ITU-R P.1546 and the publicly available noise datasets collected from the actual environment. The research shows that this algorithm can significantly suppress the influence of interference signals and environmental noise on useful signals, providing a basis for promoting the evolution of sensors towards higher reliability and robustness.

## 1. Introduction

In the context of the rapid development of the Internet of Things (IoT) and intelligent sensing technologies, sensor networks have been widely applied in environmental monitoring, industrial control, intelligent healthcare, and other fields. However, various interference signals existing in complex electromagnetic environments, such as radio frequency interference, channel noise, and multipath effects, seriously affect the signal acquisition accuracy and transmission reliability of sensor nodes. Traditional interference suppression methods, such as the iterative method with adaptive threshold (IMAT) [1,2,3], time-domain parameter interpolation [4,5], and ramp filtering (RFmin) [6,7], can achieve interference suppression in specific frequency bands or static environments, but they are difficult to adapt to the dynamic interference patterns and multi-source signal coupling characteristics in sensor networks. Especially in non-stationary noise environments or scenarios where multiple types of interference coexist, their suppression performance significantly declines.

In recent years, deep learning techniques have provided a new paradigm for interference suppression in sensor networks due to their powerful nonlinear modeling capabilities. These techniques can be classified into interference suppression technologies based on convolutional neural networks (CNN) [8,9,10,11], deep neural networks (DNN) [12,13,14], recurrent neural networks (RNN) [15,16,17], and autoencoders [18,19,20]. Among them, CNN, as an important branch of deep learning, can effectively extract spatial features in sensor signals through local perception and weight sharing mechanisms, such as the spatial correlation of multi-sensor arrays and the local structure in the time-frequency domain of signals. For instance, after converting the time-domain signals collected by sensors into time-frequency images through short-time Fourier transform or wavelet transform, CNN can automatically capture the distribution patterns of interference signals in the time-frequency domain, achieving feature separation for typical interferences such as narrowband interference and impulse interference [21,22]. However, a single CNN model has insufficient modeling capabilities for the temporal dynamic features of sensor signals, such as the duration and periodic variation patterns of interference signals, making it difficult to handle complex interference scenarios with long-term dependency characteristics.

Long Short-Term Memory (LSTM) networks, as an improved model of Recurrent Neural Networks (RNNs), can effectively capture long-term dependencies in time series by introducing gating mechanisms (input gate, forget gate, and output gate). They are suitable for analyzing the temporal evolution patterns of sensor signals. For instance, in sensor networks, LSTM can learn the evolution patterns of interference signals over time, such as the changing trends of interference intensity over time and the correlation between interference periods and environmental factors, thereby enabling the prediction and suppression of dynamic interference [23,24,25]. However, LSTM has a relatively weak ability to extract spatial features of sensor signals, such as the spatial correlation among multi-node signals and the distribution of signals in multi-dimensional feature spaces, making it difficult to fully utilize the spatial diversity advantages of sensor arrays.

To integrate the spatial feature extraction capability of CNN and the temporal dynamic modeling ability of LSTM, this paper proposes a CNN-LSTM hybrid interference suppression algorithm suitable for sensor networks. This algorithm first uses CNN to extract spatial features from multi-dimensional signals collected by multiple sensor nodes (such as acceleration signals from vibration sensors and field strength signals from electromagnetic sensors), capturing the local structural features of interference signals in the time-frequency domain and spatial domain; then, it uses LSTM to model the temporal sequence of the features output by CNN, analyzing the long-term evolution law of interference signals; ultimately, through end-to-end network training, it achieves adaptive suppression of complex interference in sensor signals. Compared with traditional methods, this algorithm does not rely on prior knowledge of interference signals (such as power spectral density and modulation mode) and can directly learn interference features from sensor measurement data, making it suitable for multi-modal interference suppression in non-stationary and non-Gaussian noise environments.

## 2. Materials and Methods

### 2.1. Optimization of Two LSTM Model-Based Interference Suppression Algorithms

The LSTM model has a certain inhibitory effect on interference signals. Based on the LSTM model, this paper optimizes its algorithm and adopts the CNN-LSTM model to suppress interference signals. The performance is quantitatively evaluated using MSE, RMSE, and regression fit. Under the same parameter conditions, a quantitative comparison is made with the interference signal suppression algorithms of LSTM, BO-LSTM, and CNN-LSTM models. Finally, the effectiveness of the algorithm was verified by using the signal propagation model based on ITU-R P.1546 and the publicly available noise datasets (babble and volvo) collected in reality. The experimental content is shown in Figure 1.

(1)The CNN-LSTM model incorporates the CNN algorithm before the LSTM prediction. Due to the unique feature extraction capability of CNN, the prediction of the signal will be more accurate.(2)The Bayesian Optimization LSTM (BO-LSTM) model uses the Bayesian algorithm, sets the objective function, and adjusts the hyperparameters of LSTM under the condition of obtaining the minimum value of the objective function, making the prediction output of LSTM closer to the undisturbed pure signal.

### 2.2. CNN Algorithm

Convolutional Neural Network (CNN) is a type of neural network specifically designed to handle data with grid-like structures, such as time series data and image data. It is an improvement over traditional neural networks, both of which adopt a layered network structure. Essentially, it is a mapping from input to output, capable of learning a large number of mapping relationships. A CNN consists of an input layer, an output layer, and multiple hidden layers, which can be classified into convolutional layers, pooling layers, ReLU layers, and fully connected layers. The greatest advantage of a CNN lies in its unique weight-sharing structure, which reduces the complexity of the network and enables it to handle high-dimensional data with ease.

### 2.3. CNN-LSTM Model

The CNN-LSTM model was originally called the Long-term Recurrent Convolutional Network (LRCN) model. Here, the more general term CNN-LSTM is used to represent LRCN. The CNN-LSTM model uses a convolutional neural network (CNN) to extract features from the input data and then uses a long short-term memory network (LSTM) to predict the output data.

The input of the model algorithm is a set of raw data, which contains interference signals. The output is the predicted value of the model. However, the training data uses pure signals without interference. The input of the CNN-LSTM network is a set of data composed of one-dimensional vectors. The parameters of the created CNN-LSTM network model are as follows:(1)One-dimensional Convolutional Neural Network: The CNN network consists of convolutional layers and pooling layers, each layer having convolutional kernels and pooling kernels, with the size set by a dimension vector; the activation functions are all ReLu or eLu.(2)Long Short-Term Memory Neural Network: The LSTM network consists of layer units, and the number of hidden neurons in each layer is set accordingly; the activation function can be chosen as ReLu or eLu.(3)Fully connected layer: The output layer of the CNN-LSTM network model, which adopts a deep neural network with a single hidden layer, and its output result is the predicted value.

As shown in Figure 2, the complete CNN-LSTM model architecture is presented. It can be observed from the figure that the model mainly consists of two key parts: Firstly, the input data is sent into the convolutional neural network (CNN), where features are extracted through convolution and pooling operations, and the dimensionality reduction in the data is accomplished. Then, the feature data processed by the CNN is passed to the long short-term memory network (LSTM). In the LSTM network, the parameters of the forget gate, input gate, and output gate are adjusted through repeated iterative training with a large amount of data, thereby learning the temporal dependencies between the features extracted by the CNN and achieving effective dynamic modeling of time series data input and output. Finally, the results obtained after training the CNN-LSTM network are passed to the fully connected layer and transformed into predicted values for output. The entire prediction process is based on training the model with training data to determine the parameter configuration of the network. The specific architecture parameters of CNN-LSTM are shown in Table 1.

### 2.4. BO-LSTM Model

The BO-LSTM algorithm, employed for comparative analysis with the CNN-LSTM algorithm proposed in this study, incorporates Bayesian Optimization (BO) to optimize the hyperparameters of the LSTM model. These hyperparameters include the historical regression length, number of hidden layers, number of hidden layer units, unit dropout rate, and initial learning rate. This approach effectively addresses the issue of model determination, thereby enhancing the accuracy of time series forecasting.

### 2.5. BO-LSTM Model Structure

The model structure diagram of Bayesian optimization is shown in Figure 3. Its calculation requires the following several steps.

(1)Create time series data, and set the time series variable as a row vector.(2)Determine the test set. Here, 60% of the sampled data is taken as the test set, and the rest of the data is used as the training set simultaneously.(3)Set the optimizer. Determine the parameters of the LSTM model, and select the parameters: the number of hidden layers, the number of hidden layer units, the regularization rate, the initial learning rate, and whether to use bidirectional LSTM. The parameter ranges are shown in Table 2, so that the Bayesian algorithm can optimize within these parameter ranges. The maximum number of iterations is set to 10 steps.(4)Call the custom objective function and use the Bayesian algorithm to optimize the hyperparameters of the LSTM model. This is the core of Bayesian optimization.(5)Read and identify the hyperparameters corresponding to the optimal training result, and simultaneously read the LSTM model trained under the optimal hyperparameters.(6)Evaluate the errors of the training set and the test set based on the best hyperparameters.(7)Make predictions on the test set.(8)Conduct error analysis.

### 2.6. The Output of the Predicted Signal

We set the experimental signal as R(t) = S(t) + J(t) + N(t), where S(t) is the pure signal, J(t) is the interference signal, and N(t) is the noise. For the experimental signal, 60% of the data is used for training. After extracting feature values through the CNN model, the LSTM performs time series modeling from the time step dimension. The remaining 40% of the data is used for testing, and the fully connected layer outputs the predicted signal Y-Pred. The key step in the training process is that the X-Train of the training set uses 60% of the signal, while the output Y-Train uses the pure signal without interference. As shown in Figure 4.

### 2.7. Regression Fitting Evaluation

The coefficient of determination, denoted as R^2^, is commonly employed in regression models to quantify the degree of concordance between predicted and observed values, with its maximum value being 1. As the R^2^ value approaches unity, it indicates a superior fit of the regression line to the observed data; conversely, a diminishing R^2^ value signifies an inferior fit. The standard computational formula is presented as Equation (1).(1)$$R^{2}\left(y,\hat{y}\right)=1-\sum_{i=0}^{n_{samples}-1}{\left(y_{i}-{\hat{y}}_{i}\right)}^{2}/\sum_{i=0}^{n_{samples}-1}{\left(y_{i}-\bar{y}\right)}^{2}$$

In the formula, 

$R^{2}\left(y,\hat{y}\right)$ represents the coefficient of determination calculated based on the true value y and the predicted value ${\hat{y}}_{i}$, which is the final evaluation metric to be obtained.

$y_{i}$ is the true value of the i-th sample that reflects the actual observed situation of the data.

${\hat{y}}_{i}$ is the predicted value of the model for the i-th sample that is the output result of the model.

$\bar{y}$ is the mean of all the true values of the samples that reflects the central tendency of the true values.

$\sum_{i=0}^{n_{samples}-1}{\left(y_{i}-{\hat{y}}_{i}\right)}^{2}$ is the Residual Sum of Squares (RSS) that measures the degree of deviation between the predicted value and the true value.

$\sum_{i=0}^{n_{samples}-1}{\left(y_{i}-\bar{y}\right)}^{2}$ is the Total Sum of Squares (TSS) that measures the degree of dispersion of the true value relative to its mean.

### 2.8. SNR, SJR, and SINR

Three values will be used in the research, SNR, SJR, and SINR, which are defined as follows:

SNR: The ratio of the power of the useful signal to the power of the noise.

SJR: The ratio of the power of the useful signal to that of the interfering signal.

SINR: The ratio of the power of the useful signal to the sum of the power of the interfering signal and noise.

## 3. Results

### 3.1. Model of Interference Signal

The original signal S(t) adopts a BPSK signal. Under the condition of SJR = 10 dB, SNR = 0:5:20 dB, the Gaussian noise is N(t), and the interference signal adopts the cosine function $J(t)=A\cos\left(2\pi f_{c}t\right)$. Finally, the input signal is R(t) = S(t) + J(t) + N(t). The specific parameters of the input signal are shown in Table 3.

The waveform of the interference-free signal S(t) is depicted in Figure 5a. Under the conditions of SNR = 10 dB and SJR = 10 dB, the waveform after noise addition is illustrated in Figure 5b, while the waveform of signal R(t) is shown in Figure 5c. Similarly, under the conditions of SNR = 5 dB and SJR = 10 dB, the waveform after noise addition is presented in Figure 5d, and the waveform of signal R(t) is displayed in Figure 5e.

### 3.2. Experimental Results of CNN-LSTM Model

#### 3.2.1. The Transmission of Predictive Signals

For the interfering signal R(t), 60% of the data is used for training, and the remaining 40% is used for testing. The batch sample size is 12, the maximum number of iterations is 20, and the learning rate is 0.005. Under the condition of a signal-to-noise ratio (SNR) of 10 decibels, all data and prediction results are shown in Figure 6.

Under the condition of SNR = 10 dB, the comparison of the predicted output value with the test signal and the pure signal without interference is shown in Figure 7.

It can be seen from Figure 7 that the predicted values not only remove some interference signals but also suppress the influence of Gaussian white noise on the pure signal to a certain extent.

#### 3.2.2. Prediction Error (MSE and RMSE) of the CNN-LSTM Model

The error is equal to the difference between the predicted value and the pure signal. Under the condition of SNR = 10 dB, the MSE of the CNN-LSTM algorithm is 0.086, the RMSE is 0.29, and the NRMSE is 0.29. As shown in Figure 8, the red dots in the red area indicate that the absolute value of the error is less than 1, while the blue dots indicate that the absolute value of the error is greater than 1.

When the SNR value range is 0–20 dB, the mean square error and root mean square error of the CNN-LSTM algorithm are shown in Figure 9.

As can be seen from Figure 9, as the signal-to-noise ratio increases, both the mean square error and the root mean square error between the predicted values obtained by the CNN-LSTM model algorithm and the undisturbed communication signal are gradually decreasing.

#### 3.2.3. Evaluation of CNN-LSTM Prediction Regression Fitting

In the CNN-LSTM model, the degree of conformity between the predicted values and the actual values was evaluated. The fitting results are shown in Figure 10.

As can be seen from Figure 10, R^2^ = 0.956. The scatter plot of the predicted values is basically distributed around the best-fit line, and the fitting degree is relatively high compared to the pure signal without interference.

### 3.3. Experimental Results of BO-LSTM Model

#### 3.3.1. Minimum Objective Function Value

During the model training process, the Bayesian algorithm can predict the position of the next optimal point based on the current historical results, then determine the parameters and conduct training to obtain the results. This operation is repeated in a loop. Under the condition of SNR = 10 dB, the objective function of the iterative process is shown in Figure 11.

As can be seen from Figure 11, the observed minimum target value remained almost unchanged from the start of the 7th iteration to the end of the 10th iteration. Among them, the algorithm reached the maximum target evaluation 10 times, and the total number of function calculations was 10 times.

#### 3.3.2. The Output Values of BO-LSTM and the Expected Values

The simulation diagram of the predicted output under the condition of SNR = 10 dB is shown in Figure 12.

As can be seen from Figure 12, the predicted output value suppresses the interference signal and reduces the influence of Gaussian noise, being very close to the pure signal.

#### 3.3.3. MSE and RMSE of BO-LSTM

The error is equal to the difference between the predicted value and the pure signal. When the SNR value range is 0–20 dB, the mean square error and root mean square error of the BO-LSTM algorithm are shown in Figure 13.

As can be seen from Figure 13, with the increase in the signal-to-noise ratio, the MSE and RMSE of the Bayesian optimization-based LSTM interference suppression algorithm gradually decrease. Especially when the SNR is around 10 dB or higher, the MSE of the BO-LSTM algorithm is less than 0.1, indicating a good performance in interference suppression.

#### 3.3.4. BO-LSTM Prediction Regression Fitting Evaluation

The regression fitting of the algorithm under SNR = 15 dB is shown in Figure 14.

As can be seen from Figure 14, when SNR = 10 dB, R^2^ = 0.95115, the scatter plot of the predicted values is basically distributed around the best-fit line, and the fitting degree with the pure signal without interference is relatively high.

### 3.4. Comparison of Four Model Algorithms (LSTM, BO-LSTM, CNN-GRU, CNN-LSTM)

#### 3.4.1. Algorithm Error Comparison

Under the same conditions, when SNR = 0 dB, the errors of the four models are compared. The red dots represent the absolute value of the error less than 1, and the blue dots represent the absolute value of the error greater than 1. The predicted output waveform and the error scatter distribution are shown in Figure 15.

As can be seen from Figure 15, under the condition of SNR = 0 dB, the mean square error of the unoptimized LSTM model is 0.56. By using the BO-LSTM model, the mean square error is 0.51, reducing the algorithm error by 5%. By using the CNN-GRU model, the mean square error is 0.33, reducing the algorithm error by 18%. By using the CNN-LSTM model, the mean square error is 0.31, reducing the algorithm error by 20%. The predicted output values obtained by the unoptimized LSTM algorithm differ significantly from the target pure signal. The waveform output errors of the CNN-GRU algorithm and the CNN-LSTM algorithm are comparable, but the CNN-LSTM algorithm has the smallest error.

The errors of the four algorithms under different SNR conditions are shown in Figure 16 and Figure 17.

It can be seen from Figure 16 and Figure 17 that the three algorithms, BO-LSTM, CNN-GRU, and CNN-LSTM, can enhance the interference suppression performance of LSTM. When SNR ≥ 0 dB, the errors of CNN-LSTM and CNN-GRU algorithms are not significantly different, and their performances are comparable. The mean square error value of the unoptimized LSTM is obviously the largest.

#### 3.4.2. Algorithmic Fitting Degree Comparison

Under the same conditions, the comparison of the prediction fitting degrees of the four algorithms is shown in Figure 18.

As can be seen from Figure 18, when SNR ≥ 5 dB, the CNN-LSTM, CNN-GRU, and BO-LSTM algorithms are significantly superior to the LSTM algorithm. And from the overall trend, the performance of the CNN-LSTM algorithm is better than that of the BO-LSTM algorithm, which is attributed to the feature extraction ability of the CNN.

#### 3.4.3. Algorithm Computational Overhead Comparison

The test was conducted under the hardware environment of a processor: Intel(R) Core (TM) i7-10510U CPU @ 1.80 GHz and 16.0 GB of onboard RAM. The average time obtained from five independent runs of the software model is shown in Figure 19.

In terms of running speed, LSTM < CNN-GRU < CNN-LSTM < BO-LSTM. The approximate running time of BO-LSTM is 1120 s, which is related to the fact that the model needs to calculate the minimum target value, and the number of function calculations is 10, which consumes a lot of time. On the other hand, LSTM, thanks to its simple structure, has the fastest running speed. The RNN model has one less gate structure than the LSTM model, so the running time of CNN-GRU is slightly faster than that of CNN-LSTM.

### 3.5. Algorithm Verification Based on ITU-R P.1546 Signal Propagation Model

ITU-R P.1546 is a standard formulated by the Radiocommunication Sector of the International Telecommunication Union (ITU-R), with the full title of “Short-range prediction method for local area communication services in the frequency range 30 MHz to 100 GHz”. This standard provides path loss prediction models for short-range communication scenarios (typically ranging from tens of meters to several kilometers) under different environments (such as urban, suburban, and rural areas) and frequencies (30 MHz to 100 GHz). It is applicable to scenarios such as wireless local area networks (WLAN), Internet of Things (IoT), and short-range wireless access, offering standardized path loss calculation methods for wireless system design, link budget analysis, and coverage assessment.

#### 3.5.1. Composition of the Core Modules of the Model

The model generates BPSK signals based on the ITU-R P.1546 large-scale propagation model and adds AWGN noise. The core module composition and functions are shown in Figure 20.

#### 3.5.2. Simulation Parameter Settings

The key indicators of the model include path loss, signal-to-noise ratio, etc. The simulation configuration parameters include frequency, distance, environment, etc. The specific settings are shown in Table 4.

#### 3.5.3. Model Data Visualization

The baseband signal is a rectangular pulse signal. After carrier modulation, the modulated signal S(t) is obtained. The path loss is calculated using the ITU-R P.1546 model, and the power of the received signal is calculated. Finally, Gaussian white noise is added to obtain the received signal R(t). The data visualization is shown in Figure 21. The visualization results demonstrate the waveforms and power spectrum characteristics of the signal at different stages.

#### 3.5.4. MSE and RMSR

The test signal is R(t), and the BPSK signal after modulation is S(t). R(t) is taken as the test value and normalized within the range of [−1, 1]; through the trained model, the predicted value Y-Pred is obtained, and the error is the difference between Y-Pred and the corresponding S(t). The LSTM output is continuous values. With an error threshold set at 0.15, the results are converted into 0/1 binary form to calculate the BER value. The error is shown in Figure 22. For this model, MSE = 0.12, RMSE = 0.35, NRMSE = 0.41, and BER = 0.039.

### 3.6. Public Noisy Dataset Validation

#### 3.6.1. Acquisition of the Datasets

The datasets consist of different noise data measured on-site in the UK by the Speech Research Unit (SRU) of the Netherlands TNO Perceptual Research Institute, with project number 2589-SAM. Each file in the datasets is 235 s long, and the data was collected using a sampling rate of 19.98 kHz, a 16-bit analog-to-digital converter (A/D), and an anti-aliasing filter, without a pre-emphasis stage. We selected two items from the noise datasets to verify the algorithm. The datasets can be obtained from the following website: http://spib.linse.ufsc.br/index.html (accessed on 18 July 2025). The data collection method of these datasets is shown in Table 5.

#### 3.6.2. Data Visualization

The verification data signal is R(t) = S(t) + J(t) + N(t), where S(t) is a pure BPSK signal, and the babble datasets or volvo datasets are adopted. N(t) is Gaussian noise. Before entering the model, the data needs to be preprocessed. The processed data is calculated to have SNR = 10 dB, SJR = 9.87 dB, and SINR = 6.93 dB. The following experiments verify the effectiveness of the CNN-LSTM model using the babble datasets or volvo datasets. The data visualization is shown in Figure 23.

#### 3.6.3. MSE and RMSR

The values of MSE and RMSE of the data are shown in the figure. The LSTM output is a continuous value. By setting the error threshold to 0.4 and converting the result into 0/1 binary form, the BER value can be calculated. As shown in Figure 24, for the babble datasets, MSE = 0.19, RMSE = 0.44, NRMSE = 0.41, and BER = 0.31. As shown in Figure 25, for the volvo datasets, MSE = 0.21, RMSE = 0.45, NRMSE = 0.43, and BER = 0.295.

#### 3.6.4. Goodness of Fit

Figure 26a shows the goodness of fit for the babble datasets, and Figure 26b shows the goodness of fit for the volvo datasets. It can be seen from the figures that the goodness of fit for the babble datasets is 0.91, while that for the volvo datasets is 0.905.

## 4. Discussion

The interference suppression technology adopted in this paper is based on the LSTM model. LSTM is good at capturing long-term dependencies in time series data and is suitable for processing signal data with time characteristics. However, relying solely on LSTM cannot fully extract the complex spatial features in the input signal, which to some extent limits its performance. To solve this problem, this paper proposes a hybrid model structure of CNN-LSTM. CNN has strong feature extraction capabilities. By embedding CNN into the overall framework, it performs multi-level feature extraction on the input signal and generates more representative feature vectors. Then these preprocessed feature values are passed to the LSTM module to analyze the time dependencies and output the predicted signal. This division of labor design not only gives full play to the advantages of the two network structures but also significantly enhances the reliability of the entire system.

From the comparison of the four models, it can be seen that the LSTM model is the simplest and has a relatively low operational cost. However, its MSE and RMSE for suppressing interference signals are relatively high. Under certain conditions, such as when SNR = 5 dB, the BO-LSTM model has better MSE and RMSE results. However, the BO-LSTM model requires iterative calculation to find the minimum target value, and when the number of iterations is 10, the running time reaches 201 s. Compared with other models, its operational cost is too high. The CNN-GRU model is similar to the CNN-LSTM model, and their algorithm results are also comparable. However, since the LSTM has structural improvements, the performance of the CNN-LSTM model is slightly better than that of the CNN-GRU model.

Although the CNN-LSTM model has demonstrated significant advantages in suppressing interference signals, there is still room for optimization. (1) The model structure can be further optimized by introducing an attention mechanism. Adding an attention mechanism module after the LSTM layer and calculating the weights of features at different time steps can enhance the model’s ability to capture key interference features, thereby improving the accuracy of local interference suppression. (2) More actual collected signals, including useful signals and various interference noises, should continue to be used to verify the generalization performance of the model method and increase its practical value.

## 5. Conclusions

This paper addresses the issue of dynamic interference affecting sensor networks in complex electromagnetic environments and proposes an interference signal suppression algorithm based on CNN-LSTM. The test signal used in the simulation experiment consists of BPSK signals, interference signals, and Gaussian noise. When SNR = 15 dB, the simulation experiment results show that the mean square error (MSE) of CNN-LSTM is 0.06, the root mean square error (RMSE) is 0.24, and the regression fitting degree R^2^ is 0.97. Compared with other LSTM, BO-LSTM, and CNN-GRU models, CNN-LSTM has the lowest MSE and RMSE, with a single run time of 58 s and relatively low operational costs. Its comprehensive anti-interference performance is significantly better. Finally, the effectiveness of the algorithm was verified using the ITU-R P.1546 signal propagation model and public noise datasets (such as the babble datasets or the volvo datasets) collected from actual environments, and the verification results were comparable to those of the simulation experiments. The research indicates that this algorithm does not require prior knowledge of interference and can effectively suppress the impact of interference signals, thereby improving communication quality. This method provides an intelligent solution for sensors to obtain high-quality data and offers a new approach for the development of highly reliable sensors.

## Figures and Tables

**Figure 1 sensors-25-05048-f001:**
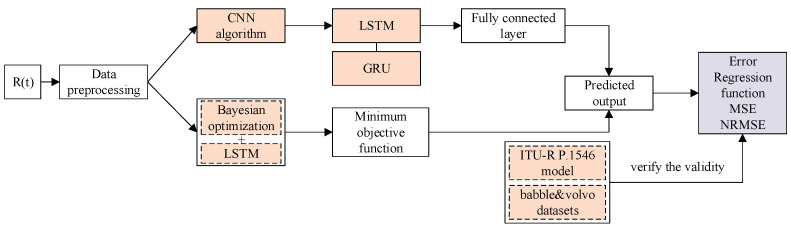
Research content framework.

**Figure 2 sensors-25-05048-f002:**
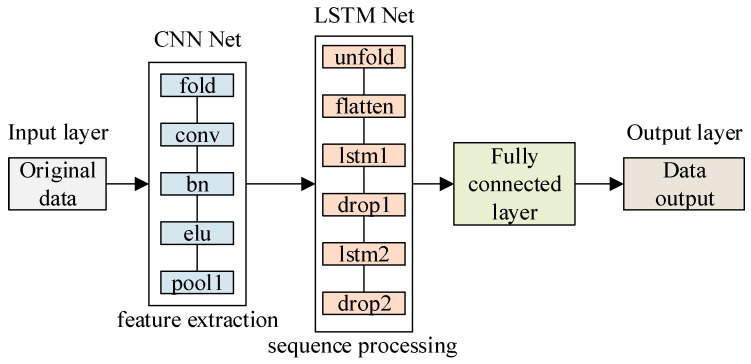
CNN-LSTM structure model.

**Figure 3 sensors-25-05048-f003:**
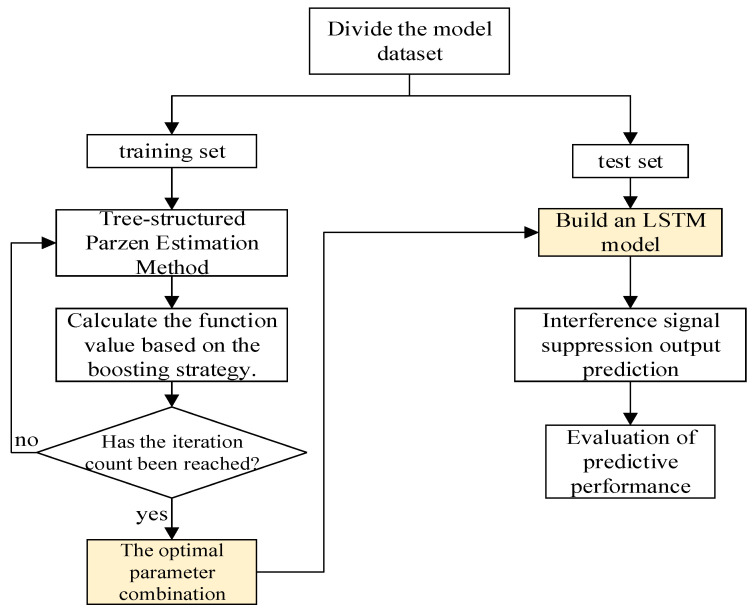
BO-LSTM model structure.

**Figure 4 sensors-25-05048-f004:**
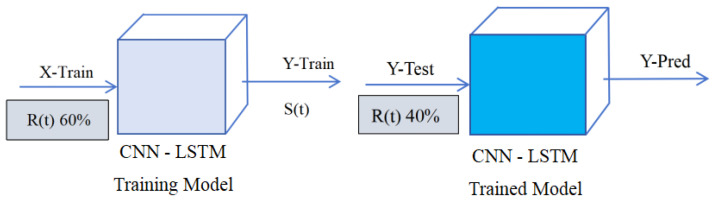
Output of the predicted signal.

**Figure 5 sensors-25-05048-f005:**
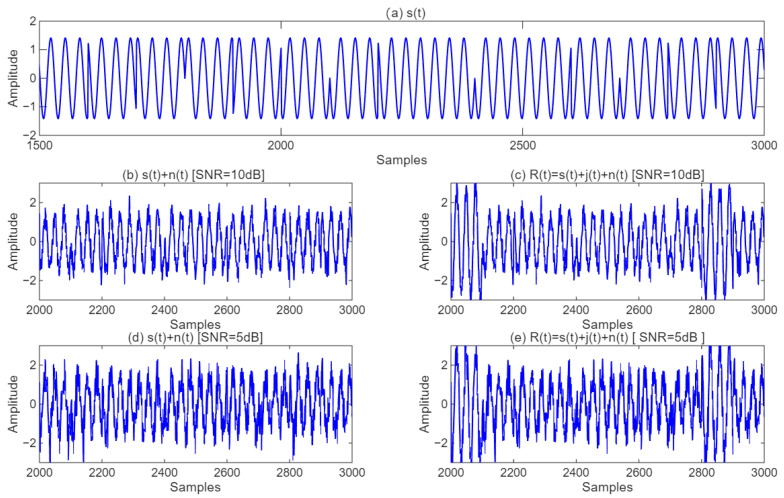
The modulus of the input signal of the CNN-LSTM model.

**Figure 6 sensors-25-05048-f006:**
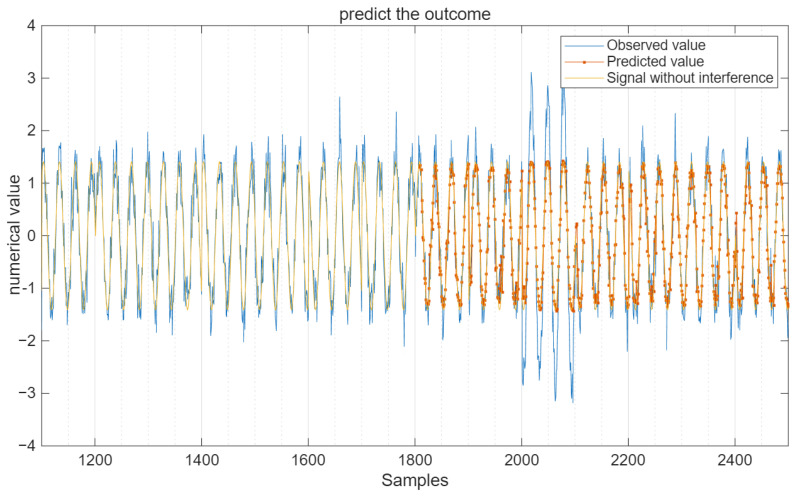
Prediction output of the CNN-LSTM model.

**Figure 7 sensors-25-05048-f007:**
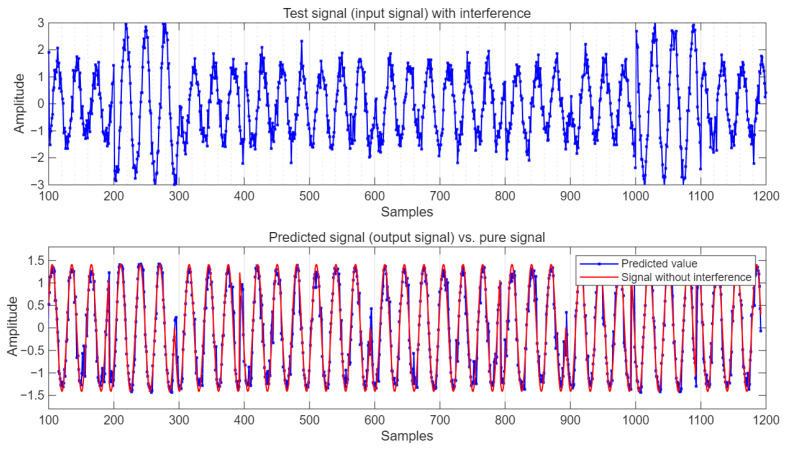
Test data and predicted output data.

**Figure 8 sensors-25-05048-f008:**
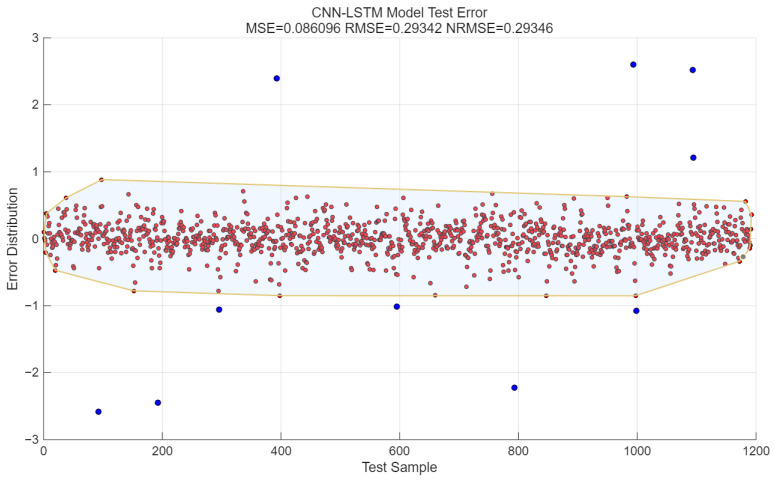
Relative error of prediction results of CNN-LSTM model. Red dots: Scatter points where the absolute value of the difference between the predicted value and the target value is less than 1. Blue dots: Scatter points where the absolute value of the difference between the predicted value and the target value is greater than 1. Yellow-bordered area: The area obtained by drawing yellow borders around all scatter points where the absolute value of the error is less than 1.

**Figure 9 sensors-25-05048-f009:**
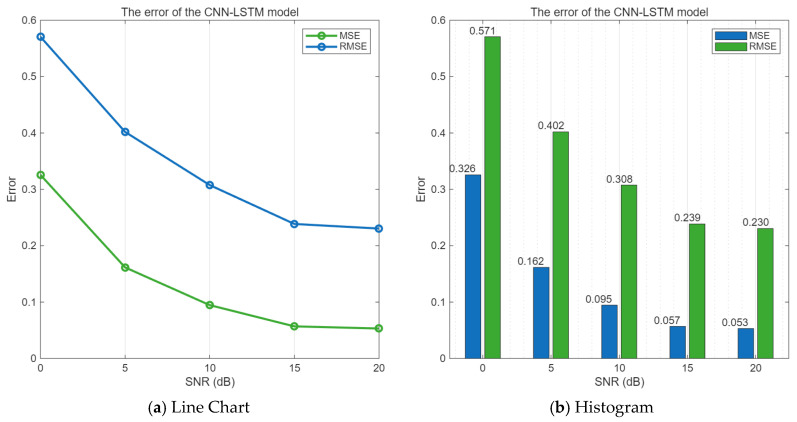
Error values of the CNN-LSTM model under different SNR conditions.

**Figure 10 sensors-25-05048-f010:**
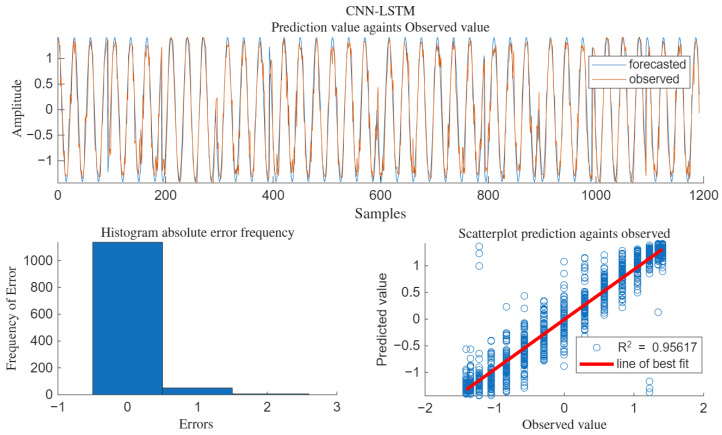
Evaluation of CNN-LSTM output value fitting.

**Figure 11 sensors-25-05048-f011:**
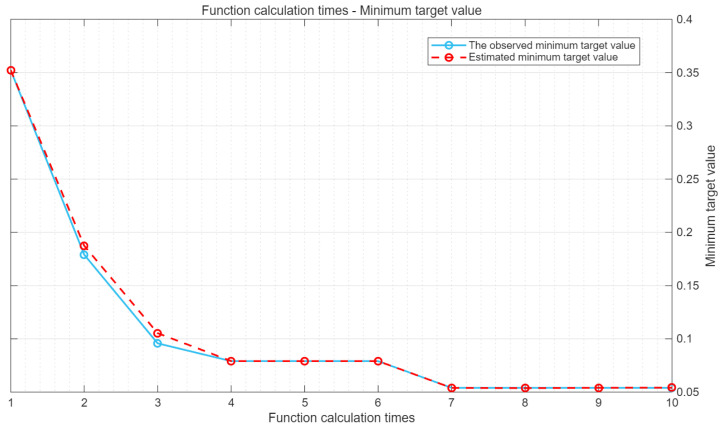
Minimum target value and function calculation times.

**Figure 12 sensors-25-05048-f012:**
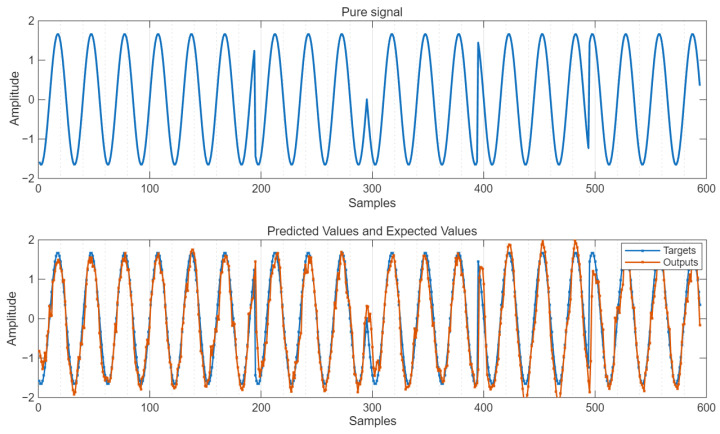
Predicted values and expected values.

**Figure 13 sensors-25-05048-f013:**
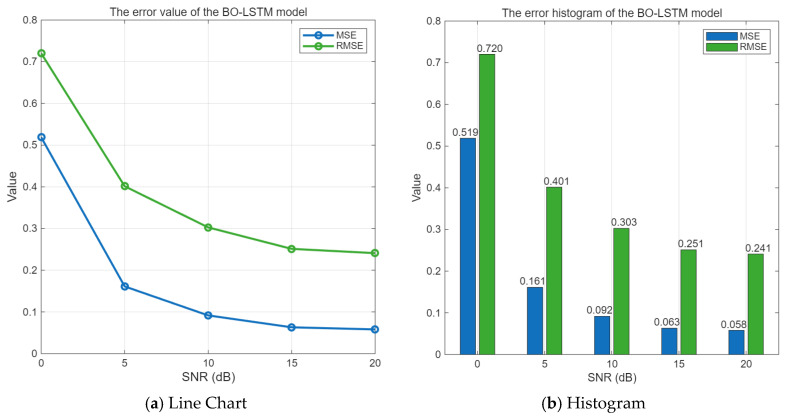
Error of BO-LSTM algorithm under different signal-to-noise ratios.

**Figure 14 sensors-25-05048-f014:**
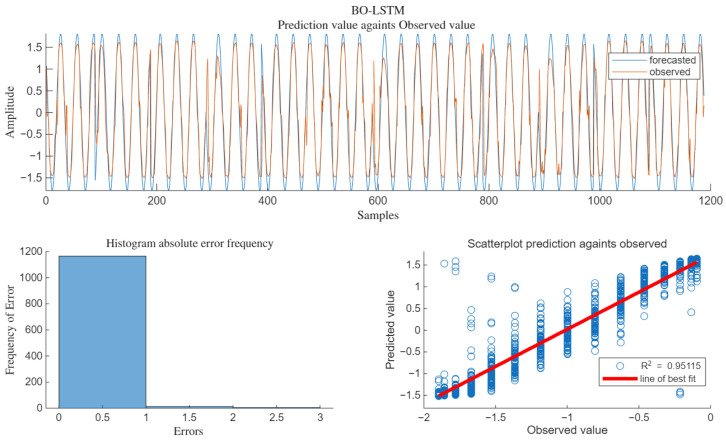
The R^2^ of BO-LSTM.

**Figure 15 sensors-25-05048-f015:**
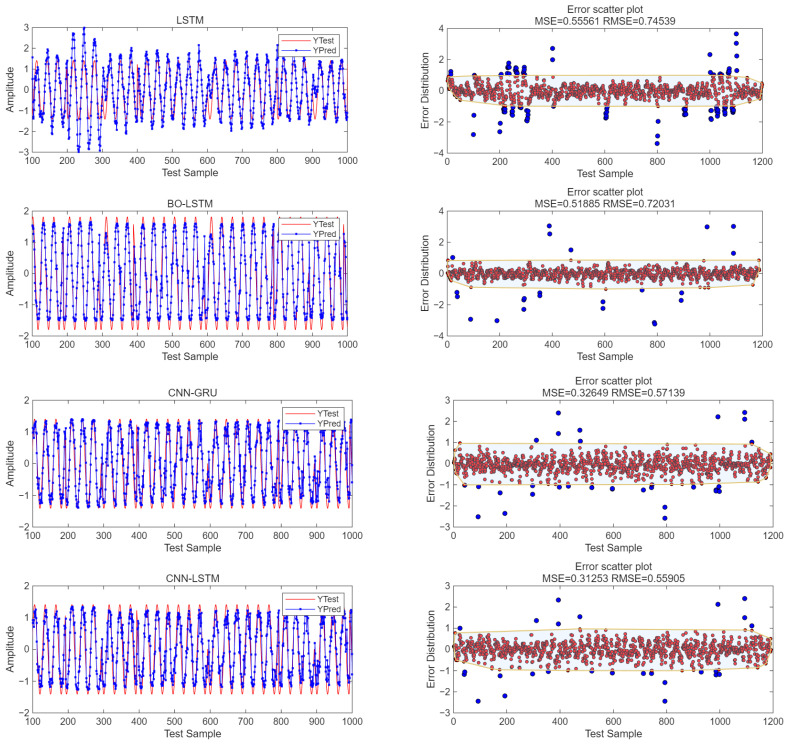
Waveform error of three algorithms. Red dots: Scatter points where the absolute value of the difference between the predicted value and the target value is less than 1. Blue dots: Scatter points where the absolute value of the difference between the predicted value and the target value is greater than 1. Yellow-bordered area: The area obtained by drawing yellow borders around all scatter points where the absolute value of the error is less than 1.

**Figure 16 sensors-25-05048-f016:**
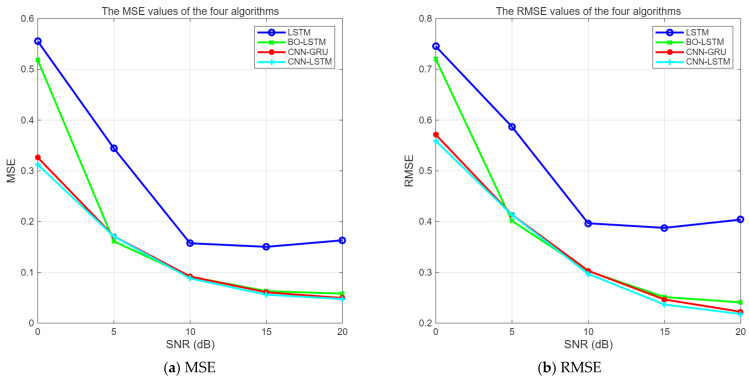
MSE and RMSE of four algorithms (line chart).

**Figure 17 sensors-25-05048-f017:**
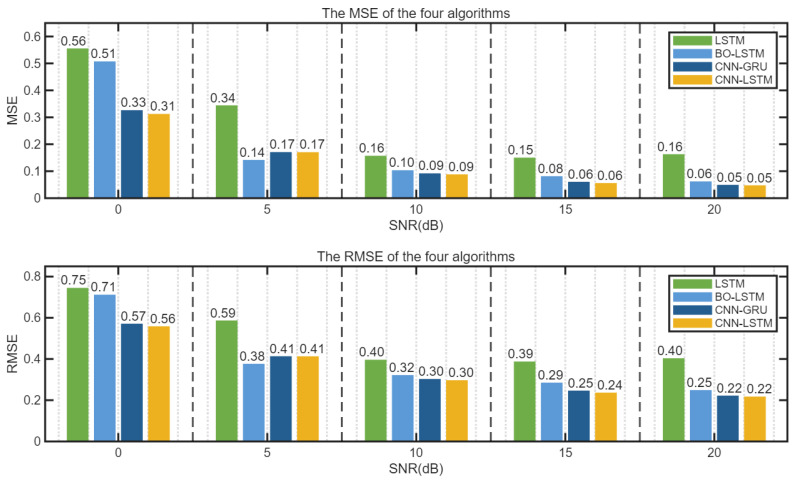
MSE and RMSE of four algorithms (histogram).

**Figure 18 sensors-25-05048-f018:**
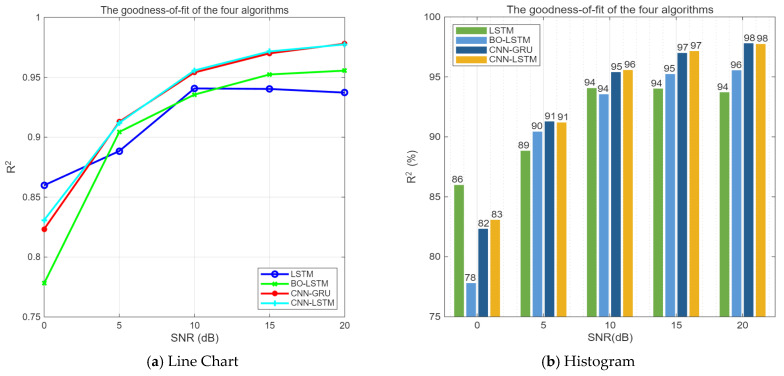
Comparison of the fitting degrees of three algorithms.

**Figure 19 sensors-25-05048-f019:**
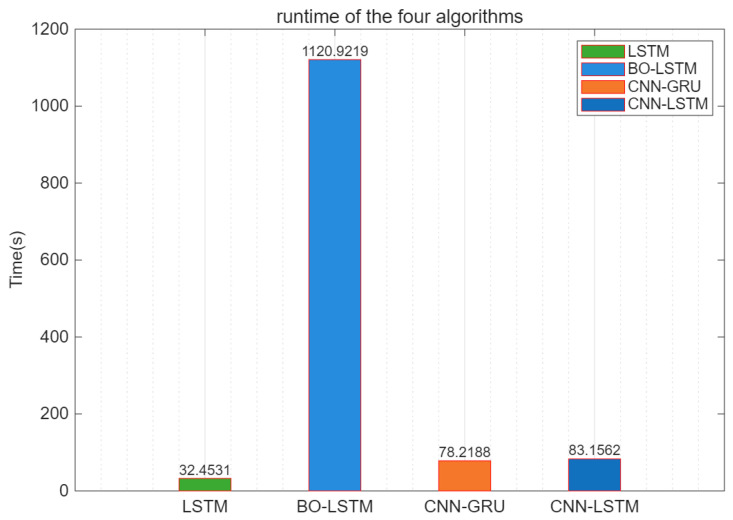
Running speeds of four algorithms.

**Figure 20 sensors-25-05048-f020:**
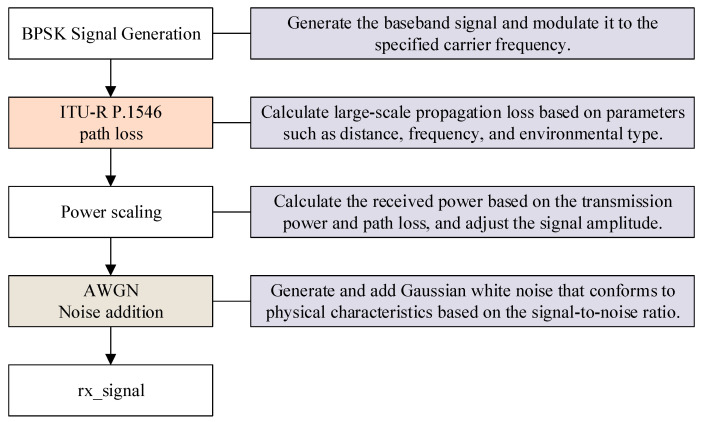
Composition and functions of model modules.

**Figure 21 sensors-25-05048-f021:**
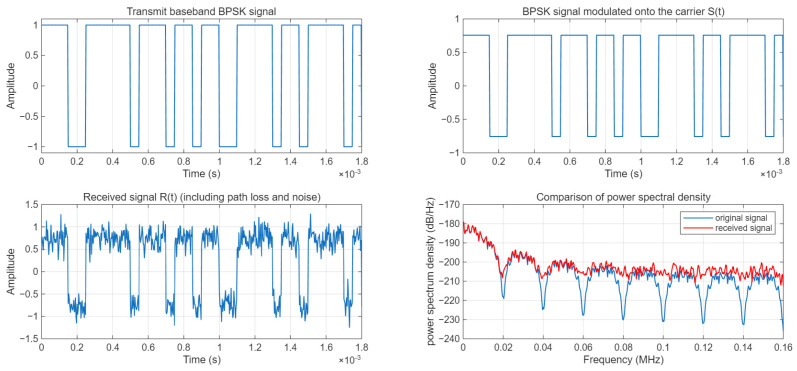
Data visualization.

**Figure 22 sensors-25-05048-f022:**
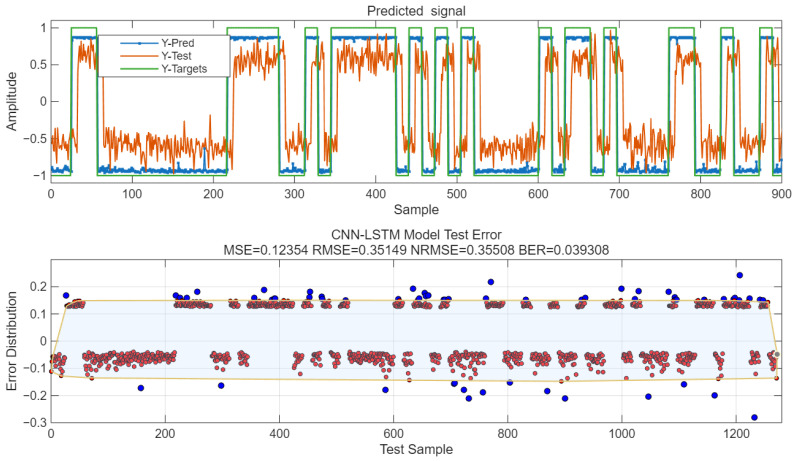
Model prediction error. Red dots: Scatter points where the absolute value of the difference between the predicted value and the target value is less than 0.15. Blue dots: Scatter points where the absolute value of the difference between the predicted value and the target val-ue is greater than 0.15. Yellow-bordered area: The area obtained by drawing yellow borders around all scatter points where the absolute value of the error is less than 0.15.

**Figure 23 sensors-25-05048-f023:**
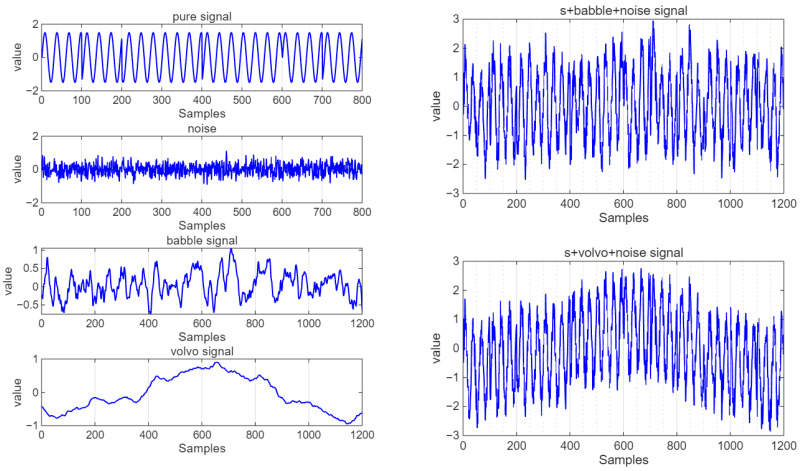
Noisy datasets.

**Figure 24 sensors-25-05048-f024:**
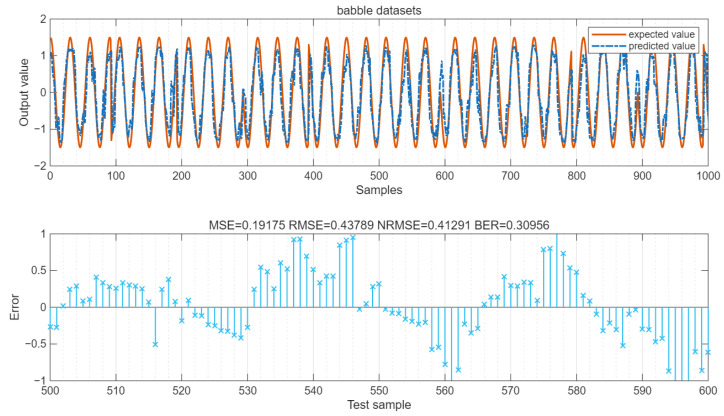
Error of the babble datasets. “×” represents the error value.

**Figure 25 sensors-25-05048-f025:**
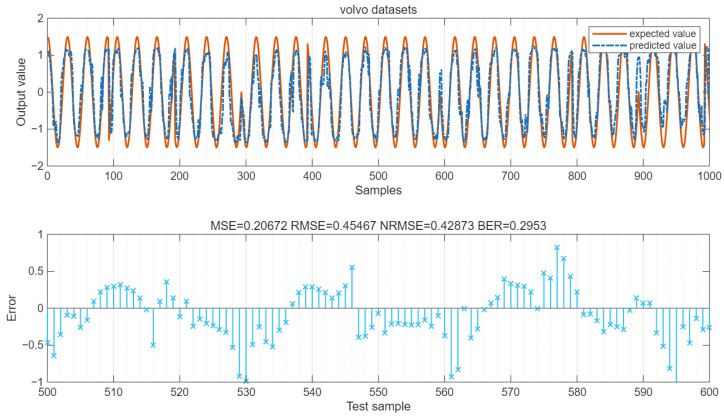
Error of the volvo datasets.

**Figure 26 sensors-25-05048-f026:**
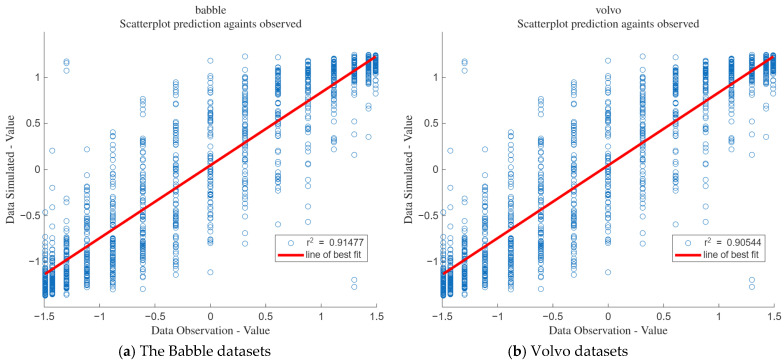
Fit degree of the noise datasets.

**Table 1 sensors-25-05048-t001:** Architecture parameters of CNN-LSTM.

Framework	Value Description
CNN layer	Filt-Size = 32
LSTM layer	2 layers. The first layer: hidden units = 128 The second layer: hidden units = 32; bidirectional propagation.
Pooling layer	1 layer, Stride = 1
dropout layer	2 layers. Drop Factor = 0.25
L2Regularization	1 × 10^−3^
Total number of trainable parameters	Approximately 1.0 × 10^5^ units

**Table 2 sensors-25-05048-t002:** Initialization parameter settings of BO-LSTM.

Framework Parameter	Value Description
Number of Layers	[1–4]
Number of Units	(50–200)
L2Regularization	[1 × 10^−10^, 1 × 10^−2^]
Initial Learn Rate	0.01–1
Use Bi-LSTM Layer	use

**Table 3 sensors-25-05048-t003:** Signal parameter values.

Parameter	Data Range
sample size	3000
signal length	3000
sample frequency	10 × 10^6^ Hz
modulation system	BPSK
SJR	10 (dB)
SNR	0:5:20 (dB)

**Table 4 sensors-25-05048-t004:** Signal propagation parameter settings.

Signal Propagation Parameters	Numerical Value
Number of symbols	200
Symbol rate	20 kbps
frequency	2.6 GHz
Transmit power	20 dBm
Transmission distance	2.5 km
environment	‘urban’
Base station height	30 m
User equipment height	1.5 m
Building density	0.3 (Effective urban environment)
Receiver noise figure	7 dB
Path loss	132.81 dB
SNR	11.18 dB

**Table 5 sensors-25-05048-t005:** Collection methods of the noise datasets.

Dataset	Recording Method	Background Environment
Speech Babble	Speech Babble acquired by recording samples from 1/2″ B&K condenser microphone onto digital audio tape (DAT).	The source of this babble is 100 people speaking in a canteen. The room radius is over two meters; therefore, individual voices are slightly audible. The sound level during the recording process was 88 dBA.
Vehicle Interior Noise (Volvo 340)	Vehicle Interior Noise (Volvo 340) acquired by recording samples from 1/2″ B&K condenser microphone onto digital audio tape (DAT).	This recording was made at 120 km/h, in 4th gear, on an asphalt road, in rainy conditions.

## Data Availability

The original contributions presented in this study are included in the article. Further inquiries can be directed to the corresponding author.

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
