# Peer review of "Interference Signal Suppression Algorithm Based on CNN-LSTM Model"

_sensors, 2025, doi:10.3390/s25165048_

Round 1
Reviewer 1 Report
Comments and Suggestions for Authors
This paper introduces a hybrid deep learning algorithm based on CNN-LSTM for interference signal suppression in wireless sensor networks. The approach leverages CNN for spatial feature extraction and LSTM for temporal modeling. It compares the CNN-LSTM model with a standard LSTM and a Bayesian Optimized LSTM (BO-LSTM) model in terms of performance metrics such as MSE, RMSE, NRMSE, and R² across varying signal-to-noise ratios (SNR). The experimental setup uses synthetic signals formed by combining BPSK-modulated signals with Gaussian noise and cosine-based interference. The results indicate that CNN-LSTM outperforms both LSTM and BO-LSTM in terms of prediction accuracy and robustness.
The concept is relevant and timely given the increasing demand for interference-resilient communication in IoT and sensor networks. The integration of CNN and LSTM is appropriate for capturing both spatial and temporal patterns in complex signal environments.
Major Comments
- The experiments are entirely based on synthetic signals (e.g., BPSK + cosine interference + Gaussian noise). The effectiveness of the proposed model in real-world noisy environments (e.g., multipath fading, non-stationary background noise) is not tested, which limits the generalizability of the findings.
- While the authors compare CNN-LSTM with LSTM and BO-LSTM, they do not benchmark it against other modern deep learning techniques such as attention-based models, autoencoders, or CNN-GRU. Inclusion of more baselines would strengthen the claims of superiority.
- The paper does not provide enough detail about how the training and test signals were generated, especially regarding the variability of the interference and noise patterns. Providing access to the code or datasets would enhance reproducibility.
- While the CNN-LSTM architecture is outlined in general terms, specific details such as the number of layers, kernel sizes, stride, dropout rates, and total trainable parameters are missing. These are critical for replicability and peer evaluation.
- The performance evaluation is primarily based on regression metrics (MSE, RMSE, R^2). However, interference suppression should ideally also be evaluated in terms of communication-specific metrics like Bit Error Rate (BER) or Signal-to-Interference-plus-Noise Ratio (SINR).
- The computational overhead of CNN-LSTM, especially for real-time applications, is not discussed. This is important for deployment in edge devices or resource-constrained environments.
Minor Comments
- There are repeated phrases across the discussion and conclusion sections. For instance, lines 400–414 in the conclusion are nearly identical to lines 376–391 in the discussion.
- The acronym "SJR" (Signal-to-Jammer Ratio) is used but not defined explicitly when first mentioned. It would be helpful to explain this alongside SNR.
- The reported NRMSE values (e.g., negative values like -7.427) are unconventional. NRMSE is typically normalized by the range or standard deviation of the target and should be a positive quantity.
- In the Bayesian Optimization section, it's unclear how the ranges of the hyperparameters were chosen and whether these were based on prior empirical knowledge or random assignment.
- Some figures (e.g., error line charts, prediction plots) are cluttered and lack sufficient resolution or labeled axes, which impairs readability.
Reviewer 2 Report
Comments and Suggestions for Authors
The paper is a mathematical exercise, based on simple simulations, on interference suppression algorithm aiming at enhance the anti-interference ability of narrow band (simulated with a cosine function) wireless communication systems through deep learning technology.
Although the authros mention radio frequency interference, channel noise, and multipath effects, which seriously affect the signal acquisition accuracy and transmission reliability of sensor nodes, in the paper they never use well-established models of radio propagation (see, for example, the ITU-R recommendations).
They never say at what frequency the sensors are working, characteristics of the antennas, terrain, receivers, etc.
In other words, the simulation is not linked to real-world scenario.
In the simulations, 60% of the data is used for training and 40% for testing. I do not see any significant improvement in these percentages.
The symbols in Eq.(1) are not defined.
Round 2
Reviewer 1 Report
Comments and Suggestions for Authors
The authors have made substantial and well-documented revisions in response to the initial review. The inclusion of real-world noise datasets, the addition of CNN-GRU comparisons, clarification of the CNN-LSTM architecture, introduction of BER and SINR metrics, and runtime analysis significantly strengthen the manuscript.
The presentation is now clear, and the methodology is transparent, including parameter ranges and dataset links. Figures have been improved, and prior concerns such as NRMSE normalization and repetitive text have been fully addressed.
Only minor editorial improvements may be beneficial but are not critical.
Reviewer 2 Report
Comments and Suggestions for Authors
The authors have partially improved the paper by introducing the noise data base used, however, they have not replied to my need of information on the wireless channel used and radio propagation environment they do mention in the Introduction.
Round 3
Reviewer 2 Report
Comments and Suggestions for Authors
The paper is now sufficiently improved.